# Health Literacy in Africa—A Scoping Review of Scientific Publications

**DOI:** 10.3390/ijerph21111456

**Published:** 2024-10-31

**Authors:** Kristine Sørensen, Verena Knoll, Neida Ramos, Millicent Boateng, Guda Alemayehu, Laura Schamberger, Stefanie Harsch

**Affiliations:** 1Global Health Literacy Academy, 8240 Risskov, Denmark; 2Pharmacoeconomics Departments, Austrian National Public Health Institute, 1010 Vienna, Austria; verena.knoll@goeg.at; 3Institute of Hygiene and Tropical Medicine, Nova University of Lisbon, 1249-008 Lisbon, Portugal; neydaneto@gmail.com; 4Ensign Global College, Akosombo 136, Ghana; millicent.boateng@ensign.edu.gh; 5University of South Africa, Pretoria 0003, South Africa; gudalem@gmail.com; 6Management Centre Innsbruck, 6020 Innsbruck, Austria; la.schamberger@mci4me.at; 7Center for Medicine and Society, University of Freiburg, 79098 Freiburg, Germany; stefanie.harsch@zmg.uni-freiburg.de

**Keywords:** health literacy, Africa, African Union, scoping review

## Abstract

Africa’s health landscape is rapidly changing, requiring new solutions such as a focus on health literacy. However, there is currently a limited overview of the development and application of health literacy in African countries and societies. This scoping review aims to analyze scientific publications on health literacy in Africa with regards to research approaches, historical developments, geographic origins, target populations and settings, and topical interests. The research followed Arksey and O’Malley six steps of scoping reviews and employed the Joanna Briggs Institute’s PCC method for search string formulation and the PRISMA-SCR checklist for reporting. On 11 July 2022, the following six databases were searched for scientific articles including included reports, policy briefs, book chapters, and research publications: PubMed, PsycINFO, Cochrane Library, ERIC, African Journals Online, and African Index Medicus. A total of 336 articles were identified. The research team co-developed a codebook and three researchers independently extracted data. The analysis provided the most comprehensive overview of the current scope and scale of health literacy in Africa to date. The publications represented 37 of the 54 African Union countries and dated back to 2001, although most were published in the last decade. The content analysis identified 13 broad themes, including mental health, communicable diseases, non-communicable diseases, maternal health, digital health, information and communication, health care, prevention and health promotion, conceptual perspectives, cultural perspectives, and outcomes and measurement. The analysis of target groups revealed a wide range of actors involved in different settings, mostly in health care or community settings. These comprehensive and novel findings can be used to prioritize future actions for public and professional capacity building, policy development, and improved practice to improve health literacy for all in Africa.

## 1. Introduction

A key driver for sustainable development is a future in which every African can enjoy a life of better health and well-being. Africa is currently the world’s second-largest and second-most-populous continent, with the fastest-growing population globally. Its population of 1.4 billion has an average age of 18.8 years, making it the youngest of any continent [1]. Sub-Saharan Africa faces considerable educational and financial challenges, with the highest illiteracy rate in the world [1] and a large proportion of people living in extreme poverty. These social determinants of health, combined with the diverse environmental and political conditions, as well as the great diversity of ethnicities, cultures, languages, and historical developments, present both richness and challenges for each African country including for the health status and health systems. Moreover, African countries face a double burden of persistently high infectious diseases and increasingly prevalent chronic diseases [2]. Nonetheless, African nations have also made remarkable strides in enhancing the health outcomes of their populations in recent decades [3]. The Africa health landscape is rapidly evolving, with better control of communicable diseases and rising prevalence of non-communicable diseases, specifically diabetes and cancer.

It is crucial to continue concerted action in Africa to curb the spread of infectious diseases, while also facilitating appropriate management of chronic conditions and general disease prevention and health promotion to support healthier populations [4]. According to the Nairobi Declaration on Health Promotion [5], investment in health literacy is key in this process.

### 1.1. Health Literacy

The concept of health literacy has received considerable attention in the past decades [6]. As the knowledge and competencies necessary to use health information and services [7], promoting health literacy has become an incremental component of many global and national health plans, e.g., the Shanghai Declaration on promoting health in the 2030 Agenda for Sustainable Development [8] or the national health literacy action plans of China [9], Germany [10], Scotland [11], and the United States [12], as an enabler of sustainable development, vital for good health and well-being, disease prevention, and achieving universal health coverage [7,13]. Health literacy is considered a relational, modifiable determinant of health that focuses on the skills of individuals and communities and the health literacy responsiveness of service providers and systems to maintain and promote health and well-being [14]. It can be defined as a multidimensional concept that includes the knowledge, motivation, and competencies for people to access, understand, appraise, and apply information to form judgments and make decisions concerning health care, disease prevention, and health promotion in everyday life to maintain and improve quality of life throughout the course of their life [6]. Importantly, it also emphasizes how health providers, organizations, and settings enable people to cope with health challenges [15].

### 1.2. Health Literacy in Africa

Although, several reports have sketched the international progress of health literacy as a growing global movement [16,17,18], they only included few records from African countries and did not provide a comprehensive overview of health literacy endeavors across the African continent. Thus, to date, there are limited insights about how health literacy is being researched, developed and implemented in the diverse African societies. Nonetheless, to improve health literacy in Africa, it is crucial to understand and address the advancements based on the distinctive African contexts [19,20].

### 1.3. Study Aim

To bridge the gap, this study aimed to shed light on the scope and scale of health literacy development in Africa by providing a first general overview of the state of the art. Therefore, the objectives included a scoping review to analyze scientific publications on health literacy in Africa with regards to (1) research approaches, (2) historical trends, (3) geographical origins, (4) target groups and settings, and (5) thematic analysis of specified health literacy aspects. The insights gained may help to inform further progress and prioritize future actions for capacity building, policy development, and informing practices on health literacy for all in Africa.

## 2. Materials and Methods

The study design followed the Arksey and O’Malley steps for scoping reviews [21] and adhered to the PRISMA Guidelines for Scoping Reviews, which allow for systematic and transparent data identification, selection, synthesis, and appraisal of research-related literature [21].

Identification of relevant studies: The search strategy employed a comprehensive and systematic approach to identify scientific evidence from publications that could potentially inform trends in health literacy development in Africa. Thus, the systematic literature search was conducted in six online scientific databases: PubMed, PsycINFO, Cochrane Library, ERIC, African Journals Online, and African Index Medicus. The search string used the PCC approach covering participants, content, and context as suggested by the Johanna Briggs Institute [22]. Thus, the search terms entailed the concept “health literacy” OR “littératie en santé” OR “compétence en matière de santé”, combined with the Boolean operator AND with the search terms for context and population “Afric*” OR “Afriq*” OR all of the names of the 54 African countries of the African Union and their associated adjectives, but NOT “African American*”. An initial search using the Portuguese term ‘literacia em saude’ did not provide any results regarding African publications and was therefore not included in the search string. The inclusion of Afrikaans, Swahili, and other languages spoken and published in Africa apart from English, French, and Portuguese was out of the scope of this study. Eventually all the six databases generated abstracts in English, thus no further analysis with regards to publication language was made. The truncation symbol was used to increase the search’s sensitivity and include additional terms, such as African countries or the African continent. Please refer to Appendix A for the search strategy. The search was not restricted to a specific time period. The study was registered at OSF https://doi.org/10.17605/OSF.IO/2QPXU (accessed on 11 September 2024).

Selection of the studies: The literature search and data collection took place in July 2022 (11 July 2022). Three researchers screened the identified records separately. Any discrepancies regarding the inclusion of studies were resolved through discussion. Articles were included which were published in English or French language and utilized the term ‘health literacy’ in the title or abstract and had full-text available online (either through open access or university access). The exclusion criteria comprised studies conducted in African regions with only limited or no international recognition, such as Somaliland and Western Sahara. Additionally, studies conducted outside of Africa, including those focusing on African Americans in the United States of America or African migrants in Europe, were excluded. Initially, the authors intended to employ forward and ancestry citation searching and manual searching of journals, yet due to the vast number of records identified and the completely voluntarily nature of this research, this proved to be beyond the scope of this research project. To ensure that no relevant studies were overlooked, the authors consulted members of the African Health Literacy Network. As critical appraisal of individual sources of evidence is not required in scoping reviews, this was not conducted.

Charting data and collating and reporting results: The lead authors KS, SH, and VK reviewed the research literature based on the pre-defined inclusion criteria. They coded the data in Microsoft Excel using a jointly developed coding scheme that reflected the research objectives such as author, year, title, country, type of manuscript, and specific study characteristics such as research methodology, target population, setting, and research theme. The data for each article was extracted by one researcher and doublechecked by two researchers. Collaboratively, they analyzed and synthesized the findings corresponding to the study objectives. Firstly, the authors extracted data based on the original wording and terminologies in the records. Secondly, the data was analyzed, main categories were developed inductively for each research objective, and the overall number of records per category was counted. We performed a qualitative content analysis of each objective to identify occurring health literacy themes and sub-themes which are all presented in aggregated forms in this article due to vast number of records found. Key informants from the African Health Literacy Network provided thorough feedback on the research design, process, and findings and contributed as co-authors to the discussion of the findings and recommendations.

## 3. Results

The findings of the scoping review of scientific publications with regards to health literacy in Africa are presented below.

### 3.1. Characteristics of the Studies

The initial search yielded a total of 876 records after removal of duplicates. The co-authoring members of the African Health Literacy Network suggested various studies, but these were already included in the initial set of 876 records. The titles and abstracts were screened for eligibility. As illustrated in the PRISMA flowchart (Figure 1), the screening process yielded 487 records, of which 137 were excluded due to inaccessibility of the full text and 13 due to unavailability of official results or inappropriateness for the purpose of the study. Ultimately, 336 records were included in the review of health literacy development in Africa. The complete list of articles can be found in Appendix A.

The scoping review and synthesis according to research methods, countries, populations, settings, and health literacy factors revealed new insights on the scope and scale of health literacy development in Africa.

### 3.2. Research Approaches Applied

Out of the 336 articles, 22 (6.5%) were scoping literature reviews, 291 (86.6%) were original research articles, and 23 (6.8%) were opinions, perspectives, or commentaries published in scientific journals or academic books. Among the original research articles, 163 publications presented quantitative methods, mostly using cross-sectional designs to assess health literacy or disease-related health literacy in patients. Other articles assessed, for example, the effectiveness of controlled trials and interventions. One hundred and five articles used qualitative approaches, with the majority using semi-structured interviews or focus group discussions to investigate barriers to health care or appropriate health behaviors. Finally, 19 publications presented mixed- or multi-method approaches, such as developing or translating health literacy measures or assessing perceptions and health behaviors related to a specific health concern, and 4 described clinical trials.

### 3.3. Publication History of African Health Literacy Research

Scientific publications on health literacy in Africa have increased exponentially since the beginning of the twenty-first century, as shown in Figure 2. The first mention of health literacy in Africa was in Kickbusch’s seminal paper “Addressing the health and education divide” [23], which provided several examples. The first specific study on health literacy was published in 2005, focusing on HIV-related health literacy in Zambia. Since then, the number of annual publications steadily increased, reaching 60 publications in 2021 and likely more in the pipeline the following years.

### 3.4. Geographical Distribution

The geographical analysis revealed that health literacy publications were available from 38 of the 54 African Union countries (Figure 3). Most studies were conducted in South Africa (*n* = 84), followed by Nigeria (*n* = 49), Ghana (*n* = 30), Ethiopia (*n* = 28), and Uganda (*n* = 27). Many countries had only one or a few studies on health literacy, such as Burundi (*n* = 1) or Sudan (*n* = 1). Additionally, 31 publications concentrated on various regions such as West Africa (*n* = 2), North Africa (*n* = 1), Southern Africa (*n* = 2), and East Africa (*n* = 1). Ten publications specifically targeted sub-Saharan Africa, while one study focused on rural Africa. Thirteen publications discussed Africa in general without specifying a country or distinct region. The publications based on general and regional accounts are not displayed in Figure 3, which only presents the country-specific studies.

For a detailed summary of the health literacy content available in each country, see Appendix A.

### 3.5. Analysis of Target Groups and Settings

The analysis of target groups distinguished between population-based and professional-stakeholder-related studies (Figure 4). The population-based studies focused on the public or subgroups of the public, while the studies focusing on the professional perspective encompassed actors in the workforce (health professionals and other relevant stakeholders), organizations, systems, or the policy arena. Figure 4 does not include the 24 publications where it was not possible to specify a target group in their research, which was the case in some commentaries or policy papers.

The largest proportion of stakeholders among all studies were patients (*n* = 80). Other subgroups included community dwellers (*n* = 33), adults (*n* = 31), students (*n* = 31), women (*n* = 23), adolescents (*n* = 23), caregivers (*n* = 18), and mothers (*n* = 15). Finally, high-risk groups, such as migrants, prisoners, the elderly, and indigenous populations, represented only 2% (*n* = 6), and several single studies focused on men (*n* = 1), consumers (*n* = 1), employees (*n* = 1), or several target groups (*n* = 2), which were grouped under “other” (*n* = 5).

Target groups associated with the healthcare workforce form the largest proportion (*n* = 39). Other professionals include professions outside the health sector (*n* = 14). A limited number of publications focused on organizations (*n* = 4) and NGOs or policy stakeholders (*n* = 3).

According to the WHO [24], a setting for health refers to a place or social context where individuals engage in daily activities, and where environmental, organizational, and personal factors interact to influence health and well-being. The findings revealed that the most common settings studies were the community and the health care settings, followed by educational settings such as secondary school or high school. As shown in Figure 4, fewer studies were designated in digital/online settings, workplaces, or rural/urban settings.

### 3.6. Thematic Analysis of Health Literacy in Africa

The thematic analysis identified four general trends covering 14 themes based on the types of health literacy categorized in the data, as illustrated in Figure 5. The trends and themes are summarized in Figure 5 and briefly described in the text with keywords and number of papers identified for each theme. It is beyond the scope of this article to provide a more detailed description based on all the records. Instead, for the sake of transparency, the general data set is provided in Appendix A.

Firstly, there was a *disease-oriented focus* concentrating on mental health, communicable diseases, and non-communicable diseases. Moreover, maternal health was a priority focus, given the high levels of child morbidity and mortality in many African countries [25].

*Mental health literacy* included 72 records related to a wide variety of themes including body image, COVID-19 and tuberculosis, suicide and depression literacy, obsessive-compulsive disorder (OCD) literacy, mental health literacy in general, mental health care and screening, mental disorders and illnesses, for instance schizophrenia, mental health governance, school mental health, socio-cultural factors, stigma, and promotion of well-being. The publications presented and discussed facets such as knowledge and capacity building, explanatory models, attitudes, impact of media, prevention and treatment, management, and policy.*Communicable diseases* included 55 records describing health literacy in a wide range of diseases such as HIV/AIDS, COVID-19, tuberculosis, malaria, Ebola, onchocerciasis, cholera, hepatitis, schistosomiasis, and foodborne diseases.*Noncommunicable diseases (NCD)* entailed 59 records describing NCD literacy in general and in relation to specific conditions such as aphasia, back pain, cancer, chronic obstructive pulmonary disease (COPD), cardiovascular diseases (CVD), epilepsy, gastrointestinal diseases, diabetes, hypertension, rheumatic heart disease, podoconiosis, systemic lupus erythematosus, and stroke.*Maternal health literacy* covered 26 records related to family planning, community care, healthy pregnancy, antenatal and postnatal care, neonatal jaundice, mortality and infant survival practices, and childcare by parents and other caregivers.

Secondly, a trend focused on *organizational and systemic aspects* of health literacy in the domains of health systems, health care, prevention, and health promotion.

5.*Health system* entailed 21 records regarding health service literacy, health system literacy, and nursing literacy.6.*Healthcare* included 11 records to health system issues such as treatment and medication literacy. Some specific features were identified, such as autopsy literacy, hemodialysis literacy, obstetric literacy, organ donation literacy, palliative care literacy, and genomic literacy.7.*Prevention* covered 24 records associated with reproductive health literacy, HIV/AIDS literacy and sexual health, and vaccine literacy related to COVID-19 and HPV. The theme also included alcohol and drug literacy as well as oral health literacy, hygiene, and antimicrobial resistance literacy. In addition, occupational health literacy and health literacy in relation to child labor were included.8.*Health promotion* included 17 records focusing broadly on health literacy-related health promotion strategies such as sexual health literacy, nutrition literacy, disability literacy, and self-care literacy, as well as environmental health literacy.

Thirdly, there was a clear focus on *communication and information*, particularly in relation to *digital health*.

9.*Information and communication* health literacy was based on 19 records and focused on awareness, seeking and accessing information, and improving communication between, e.g., patients and healthcare providers.10.*Digital health literacy* concerned seven studies and discussions on the use of eHealth literacy, mHealth literacy, social media, and other forms of technology and innovation.

Lastly, there was a conceptual focus on concepts and cultural perspectives as well as outcomes and general measurement in relation to health literacy.

11.*Conceptual perspectives* referred to seven reflections on concepts and approaches related to health literacy, for instance, functional health literacy and the life course perspective.12.*Cultural perspectives* included one study that highlighted the importance of cross-cultural understanding and collaboration.13.*Outcomes* of health literacy were based on 13 records highlighting outcomes such as health behavior and practices, health status, participation and empowerment, quality of life, sustainability, and equity.14.*Measurement* included 13 records emphasizing the wide range of tools, methods, and approaches used to assess the prevalence of health literacy or as an outcome of interventions. Tools included, for example, REALM-R, HLS-EU, HLQ, and the Mental Health Literacy Survey.

## 4. Discussion

This comprehensive scientific review provides a novel overview of the scope and scale of health literacy developments in Africa. By incorporating global and African-specific scientific databases, the literature search was rigorously conducted and provided ample evidence that health literacy is an emerging feature of interest on the African continent. The large number of 876 publications covering health literacy, of which 336 publications were included, demonstrates its importance for research, policy, and practice in Africa.

According to the study, the global community invested in health literacy, health promotion, and global health has a unique opportunity to learn successful approaches, particularly in adapting questionnaires and interventions to linguistic and cultural contexts from the African community. Over half of the research made use of quantitative approaches illustrating validation and measurement studies are common in Africa like in other parts of the world [26,27]. While no in-depth analysis of definitions was conducted, the general review of literature revealed that the use of health literacy definitions referred to the most widely used in research [6].

Although health literacy in Africa was briefly explored since 2001, a major shift in research development can be detected since 2017 where the research publications significantly increased. This trend is similar to the global trend, albeit on a smaller scale [18].

Across the African Union, South Africa, Nigeria, Ethiopia, Ghana, and Uganda lead the research development on health literacy. It is worth noting that these countries consist of comparatively large populations in Africa. Moreover, all of these countries utilize English as one of the official languages, a legacy of the colonial era when they were under British rule. The question of whether establishment of universities in these countries in the 20th century by Britain, along with special scholarship programs (such as the Commonwealth Scholarship), or other programs that initiated and intensified long-standing collaboration between universities in these countries and those in the Global North played a role in faster adoption of new developments in global health cannot be satisfactorily answered based on the available data. Interestingly, most publications originated from English-speaking countries, with limited contributions from French- or Portuguese-speaking countries. The research and publication opportunities within and between countries may be affected by a lack of funding or language barriers for non-native English speakers in relation to publishing [28].

The analysis of target groups indicated a greater emphasis on individual health literacy rather than on the organizational health literacy perspectives. This finding is consistent with global developments. This might be grounded in the historical evolvement of the concept which primarily focused on individual health literacy before embracing organizational health literacy responsiveness [6,29].

In contrast to health literacy research in other world regions, the settings analysis highlighted the interesting feature that health literacy is promoted more extensively in communities and settings outside the health sector. Studies in Europe and North America often focus on clinical settings [30,31]. The findings can be partly explained by the health challenges prevalent in certain areas of Africa, where access to health services is limited, universal health care is scarce, and health system literacy is often low [32]. Consequently, interventions located in the community might be more appropriate and many African countries are more familiar with it, as they resemble the many interventions aimed at preventing infectious diseases such as malaria or HIV [33,34].

The thematic analysis emphasized the specific health challenges facing Africa. Firstly, with regard to communicable diseases, health literacy was widely associated with communicable diseases, predominantly in the African region, such as Ebola, HIV/AIDS, and malaria [35]. Moreover, the emphasis on maternal health is due to the heightened risk of morbidity and mortality within childhood (under-5 mortality) [25]. The study suggests that mental health literacy remains a challenge due to insufficient knowledge and awareness, stigma, and prejudice, which is consistent with findings from other research studies [36]. The widespread focus on non-communicable diseases such as cancer, diabetes, and cardiovascular diseases reflects the projected increase in global burden [37]. Secondly, health information and communication remain an important avenue for raising awareness, especially through digital health literacy such as mHealth literacy, which appears to be a promising strategy given the increased access to mobile coverage [38]. These findings align with global trends and policies [39]. Thirdly, a cross-cutting topic was sexual and reproductive health literacy related to family planning, which remains a challenge in the world’s youngest population [40]. Fourthly, the topical analysis highlighted efforts to bend the curves of limited health literacy regarding health systems and organizations, health care, disease prevention, and health promotion to improve outcomes. This is a universal challenge that is prevalent around the globe [16]. Finally, the discussion of conceptual approaches, measurements, and ways to address cultural change was dominated by the academic perspectives, in line with international research developments [15]. While this scoping review aimed to provide a general overview of the scope and scale of health literacy developments in Africa, the findings show that many more lessons can be learned through a more in-depth analysis of the themes and sub-themes in the future.

### 4.1. Strengths and Limitations

Despite the rigorous study design, some limitations remained. Data search and collection focused on countries officially recognized as members of the African Union, which excluded countries and territories such as Somaliland and Western Sahara. These choices were made for administrative reasons but may have introduced bias into the data set. Moreover, the main purpose of this study was to provide a general overview of articles on health literacy trends in Africa, not to challenge and critique geographical boundaries. This study used official national borders, such as those used by the African Union, to categorize different geographic regions, recognizing that these borders do not always reflect the reality of ethnic groups that are found on either side of national borders. Given the vast diversity of ethnic groups across the African continent, including their linguistic, cultural, geographical, and often socio-economic particularities, it is crucial to account for these factors when designing future studies.

The data collection was based on six international and Africa-focused databases. While this may not be an exhaustive list of all health literacy publications, the Africa-specific databases clearly yielded studies that would not have been found elsewhere. Scanning the gray literature was beyond the scope of the study; however, to be inclusive, all types of resources identified in the databases were included based on the inclusion criteria, not just the research publications. An initial test search revealed no results for Portuguese publications; hence, the study was based on a search of key terms in English and French, which are also the only or primary languages used as interfaces for these databases. A considerable number of these databases offer abstracts in English (or French) even in instances where the original article is available in a different language. In accordance with the aforementioned strategy, the authors were thus able to guarantee that all records with an abstract in one of these languages were identified, irrespective of the language employed in the full text. Following the screening of both the abstract and the full text, only records in English remained. It is possible that the application of this search strategy may have resulted in the overlooking of records in other major languages spoken in Africa, such as Portuguese, Swahili, or Afrikaans, which have not been indexed with an English or French abstract in the databases consulted. The aim of this review was analysis of publications using the term ‘health literacy’, so other studies that indirectly address ‘health literacy’ without mentioning the concept were not included. This review presents the general results of a broad trend analysis of health literacy publications in Africa. An in-depth meta-analysis of research articles or policy recommendations was beyond the scope of this review, although the data were available. Further research and publications are warranted to fill these gaps.

### 4.2. Implications for the Future

Reviewing the developments of health literacy in Africa sparks food for thought. First and foremost, the study revealed that health literacy is on the agenda in Africa and that progress is expanding. Improving health literacy in Africa requires a multifaceted approach involving policy integration, education system enhancement, community engagement, technological innovation, health system improvements, research, and partnerships. It is paramount to create more visibility regarding the need for health literacy and good practice interventions and their importance in improving health outcomes and the quality of health services. By addressing these areas, African countries can empower their populations to make informed health decisions, ultimately improving health outcomes and reducing health disparities across the continent.

*The study highlighted the importance of contextualizing health literacy with respect to African society, culture, and traditions*. A co-author highlighted that when searching Google and consulting the scientific discourse, health literacy is often portrayed, e.g., with an image of a person sitting in front of a computer navigating for health information or discussing and negotiating treatment options with a physician. These situations may not be appropriate in many African settings, where large portions of the population are rural dwellers, have limited access to digitalization, and often low functional literacy. Thus, the strategies, tools, and approaches for health literacy should be tailored to local opportunities and needs, for example, by translating and validating tools for health literacy assessments, using social mobilization, community dialogue, and engaging local religious and community leaders to provide information about health and available health services. Drawing on local wisdom and strategies of co-design and participation is recommended to define and conceptualize health literacy for each unique African context [41]. This can be performed in collaboration with health literacy practitioners in the specific community, district, region, or country with the aim of making it a priority agenda for professionals and policymakers through advocacy using multiple platforms. Health literacy initiatives can tackle social norms through proven interventions, including stakeholder engagement and community dialogue. Harmful practices adversely affect the groups most at risk. Given the degree of vulnerability among the general population, efforts to engage the most vulnerable are often deprioritized to maximize scarce resources.

*In Africa, where health challenges are significant and resources are often limited, enhancing health literacy can play a transformative role.* Similarly to trends in other world regions, health literacy should be a core component of national health policies and strategies. Governments should recognize it as a critical determinant of health and integrate it into treatment pathways and health promotion and disease prevention programs. This includes developing resources and providing ongoing professional development opportunities for health professionals, teachers, and community workers and leaders alike. As this study shows, it is important to collaborate with community leaders, religious figures, and traditional healers to promote health literacy. These trusted figures can help disseminate health information and encourage positive health behaviors. Use of health literacy champions may pave the way for better implementation of health literacy [42].

*Local and international investment and networking are needed for further research, better interventions, and policy development to advance health literacy in Africa.* For example, health literacy strategies can be included in the health sector development plans of each African country. Investments in more health literacy interventions can benefit both populations and systems specifically if cultural sensitivity is considered when developing and implementing the interventions. The *African Health Literacy Network* is a crucial player in strengthening collaboration and further dissemination of the health literacy agenda to build the capacity of people, professionals, and societies in the African region. It was launched in 2017 and has grown to cover membership from 25 countries. It hosted its first pan-African conference in August 2023.

*Establish mechanisms to monitor and evaluate the effectiveness of health literacy programs in Africa.* This includes conducting research to identify best practices and areas for improvement. The lessons learned from this study revealed that measurement and evaluation of health literacy takes place and can be amplified more progressively. Collecting data on health literacy levels across different regions and demographic groups to inform policy decisions and tailor interventions accordingly will improve the quality of implementation. As an example, the WHO is supporting countries like Liberia and St. Tome and Principe to measure population health literacy to inform and develop the national health literacy agendas (contact the corresponding author for more information).

## 5. Conclusions

In conclusion, this review of health literacy trends in Africa underscores the call to action to develop the field of health literacy in ways that are responsive to the needs and concerns of African countries and contexts, rather than copying what has already been performed elsewhere without much adaptation. As public health services are currently being strengthened and expanded in many African countries [40], the valuable lessons learned from this study can be used as a reference point and incorporated from the outset, particularly with regard to building systemic health literacy capacity [43] and incorporating health literacy practices as part of community development.

Building on the widespread legacy of health education and community development in the African region, new approaches applied through a health literacy lens can help to target populations in vulnerable situations and make services more fit for purpose. Proposed solutions include co-designed, targeted programs and interventions for specific populations, as well as capacity building of the workforce and investment in education policies. Particular attention should be paid to ensure cultural appropriateness and adaptation to the needs and perceptions of the populations, especially women and high-risk groups, to leave no one behind.

Strengthening health literacy in Africa is essential to building a future where every African can enjoy a life of better health and well-being. Importantly, this scoping review highlights that and how health literacy development is on the rise and is promisingly being recognized as a positive key driver of change across the African continent.

## Figures and Tables

**Figure 1 ijerph-21-01456-f001:**
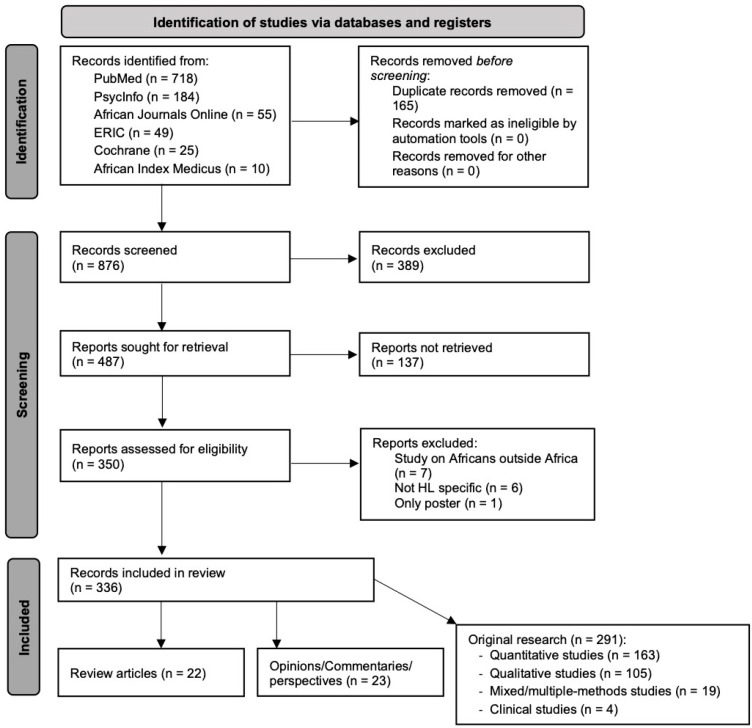
Systematic data search for the scoping review of health literacy development in Africa.

**Figure 2 ijerph-21-01456-f002:**
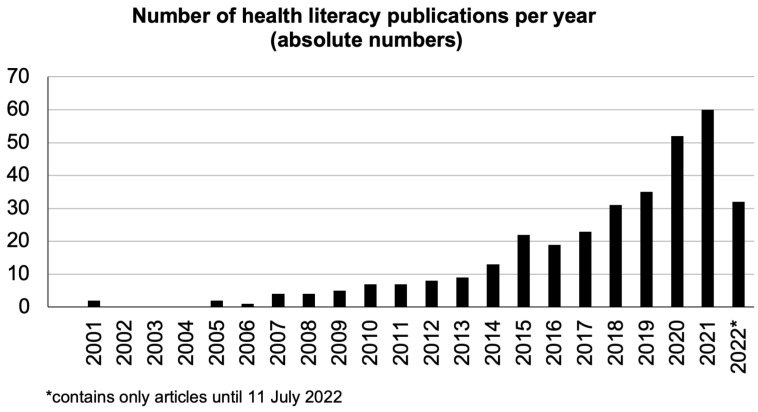
Publication history concerning health literacy in Africa.

**Figure 3 ijerph-21-01456-f003:**
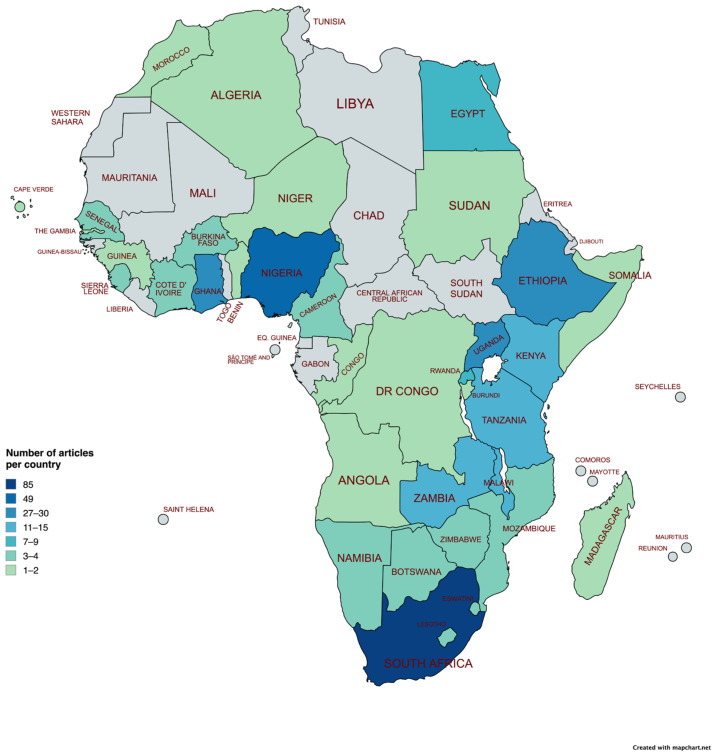
Geographical distribution of health literacy development in Africa (absolute number of publications per country identified in the review, excluding general and regional sources). Map created by authors using the program mapchart.net.

**Figure 4 ijerph-21-01456-f004:**
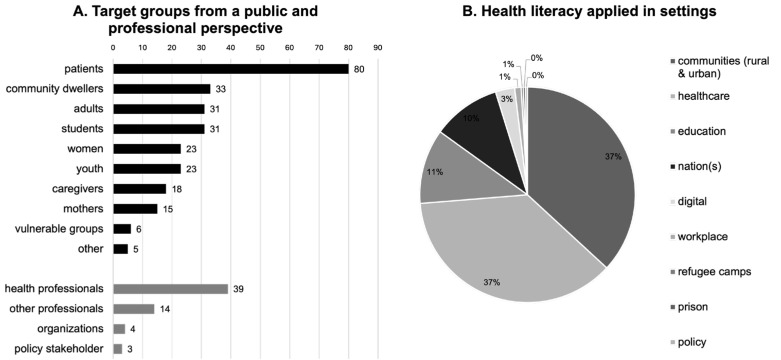
(**A**) African trends regarding health literacy target groups from a public and professional perspective. (**B**) African trends concerning health literacy applied in settings and communities.

**Figure 5 ijerph-21-01456-f005:**
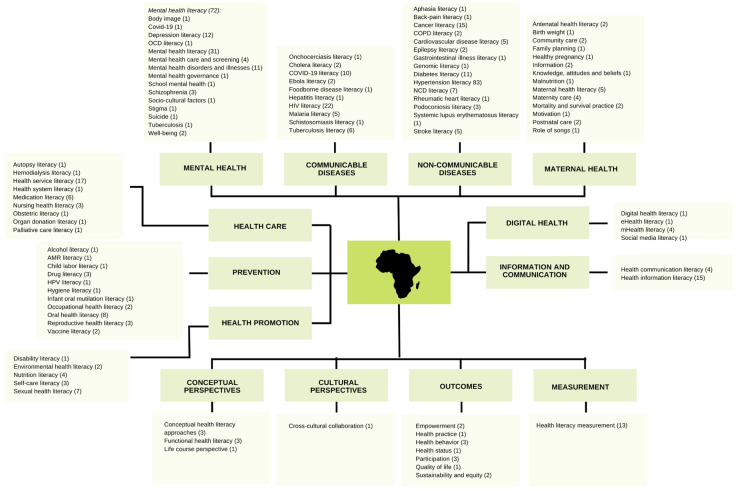
Topical analysis of health literacy developments in Africa.

## Data Availability

Dataset available on request from the authors.

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
