# Peer review of "Health Literacy in Africa—A Scoping Review of Scientific Publications"

_ijerph, 2024, doi:10.3390/ijerph21111456_

Round 1
Reviewer 1 Report
Comments and Suggestions for Authors
Dear Respectable Authors
Thank you for considering a great area of research related to health literacy in African countries. You conducted a scoping review to investigate some approaches to these issues in these countries. Your results are of interest but the way you report the manuscript needs some revisions as follows;
- Please add clear study aims to the abstract.
- Please add some details of methods in the abstract including database name, date of search, data charting, and synthesis methods.
- Lines 19-20, please add a percentage for each type of study.
- Please add the "scoping review" to the keywords.
- Methods, please add subheadings based on PRISMA-ScR and re-structure your methods section.
- The title of Appendix A is not good. Please refine it as "Search strategy".
- The title of Appendix C is not good.
- Please add the type of language used for the included studies. Please insert a column in Appendix B.
- Lines 114- 119 including Appendix B and Figure 1 are related to results, not methods. Please remove it from here and add it to the first paragraph of the results. Please follow the PRISMA-ScR.
- You stated that conducted a stakeholder analysis but there is no information regarding this in the methods section.
- Please restructure your result considering items 14- 18 of the PRISMA-ScR.
Cheers
Author Response
Thank you for the kind advice. Please see responses below.
Best regards
Kristine Sørensen
Reviewer 1.
|
- Please add clear study aims to the abstract. |
We reformulated the aim in the abstract |
|
- Please add some details of methods in the abstract including database name, date of search, data charting, and synthesis methods. |
The information has been added, however, it extends the word count beyond the abstract limitation. Advice from the editor is warranted in this case. |
|
- Lines 19-20, please add a percentage for each type of study. |
Thank you for this recommendation. We had added this information in the full text but abstained to add it in the abstract due to the word limitation |
|
- Please add the "scoping review" to the keywords. |
We have now added scoping review to the keywords |
|
- Methods, please add subheadings based on PRISMA-ScR and re-structure your methods section. |
Thank you for this recommendation, however as we followed the structure of Arksey and O’Malley we decided to add their steps as the heading (here three) instead of adding 9 subheading as the Prisma-ScR lists. All information of the Prisma-ScR is listed but we found that Arksey and O’Malley propose a better chronological structure than the checklist |
|
- The title of Appendix A is not good. Please refine it as "Search strategy". |
Are you referring to the title of the Appendix or the description of the Appendix at the end of the manuscript? (see below). We adjusted it |
|
- The title of Appendix C is not good. |
Are you referring to the title of the Appendix or the description of the Appendix C at the end of the manuscript? In our view, we could give a short description of what the appendix contains at the end of the manuscript. These are not the titles. As soon as you open the Appendix you will see that the title is different. For your convenience we added both the title and the description at the end of the manuscript. |
|
- Please add the type of language used for the included studies. Please insert a column in Appendix B. |
It is not necesseary to include an additional column as all the studies were in English language. In our view, there is no need for adding a column that contains the same information all through.
|
|
- Lines 114- 119 including Appendix B and Figure 1 are related to results, not methods. Please remove it from here and add it to the first paragraph of the results. Please follow the PRISMA-ScR. |
We changed it. |
|
- You stated that conducted a stakeholder analysis but there is no information regarding this in the methods section. |
This is a missunderstanding we identified and analysed stakeholders/target groups like we analyzed settings and research approaches. we did not perform a stakeholder analysis in a narrow sense. We changed the wording and use the word ‘target group’ throughout the paper. |
|
- Please restructure your result considering items 14- 18 of the PRISMA-ScR. |
As indicated above, we were following Arksey and O’Malley’s structure and presented data along the research questions and not along the PRISMA-ScR. The structuring along the research questions is also suggested in 17. All the relevant content requested in PRISMA-ScR is presented. As we included 336 studies we cannot provide all the citations for each of the statements made. We provide examples and further information can be provided on request. |
Reviewer 2 Report
Comments and Suggestions for Authors
The article presented is relevant to the study of health literacy in Africa. The methodology used is appropriate and makes it possible to take a general approach to a topic that still needs to be explored in greater depth. The presentation of the results is well structured, but the article could benefit from a more critical discussion of the literature found and whether, in fact, these studies are meeting the needs of promoting health literacy in Africa.
There are a few aspects that I think need improvement or clarification:
Lines 106 to 109 - explain why you only included articles in English and French; by excluding other languages widely spoken in Africa (e.g. Portuguese) you may be leaving out significant studies for the topic in question. In line 285 you mention that there is little contribution from Portuguese-speaking countries, but you didn't include that language in the study... (lines 339 - 340.) I suggest that this limitation be clearly presented and indicate what weaknesses it brings to the article.
Lines 109 to 112 - the exclusion criteria indicate that countries or regions that are not internationally recognized have been excluded, couldn't this be introducing an error into the study? and why have you done this? aren't there issues that need to be reflected on these territories? I suggest explaining the need for ‘international recognition’ more clearly than what is presented in lines 325 - 332.
Lines 121-123 - I suggest explaining ‘They coded the data in Excel using a jointly developed coding scheme that reflected the research objectives’ what did this ‘jointly developed code’ consist of?
Lines 138 - 143 - You should elaborate on how you analyzed the qualitative studies. As well as reflect on the constraints of relying primarily on quantitative studies (lines 273 - 274).
Lines 155 - 163 - I suggest that a critical analysis should be made of the results obtained with regard to geographical distribution, can you reflect on the dominance found in studies carried out in South Africa and Nigeria, what factors could be responsible for this? academic factors? greater perception of the importance of health literacy? (more in-depth on lines 282-283)
Lines 172 - 173 - isn't this a weakness? Why did you include these types of articles?
Author Response
Thank you for the fruitful feedback. Please find our comments here.
Best regards
Kristine Sørensen
Reviewer 2
|
1. Please indicate whether a protocol exists for this review and if so, where. (See PRISMA guidelines.) |
Yes, we did and we added the link. |
|
2. You have mentioned that the search was done across six online scientific databases. Did the authors consult a librarian to set up the search, or is one among the team members an expert in information science? If so, it should be mentioned. |
No, in this case we did not consult a librarian but the principal investigators KS and SH are experienced researchers who have a track record of conducting scoping reviews in the field of public health and health literacy and this review provided more lessons learned. However, the point of including advice from a librarian is well received and will be applied in future studies. |
|
3. How many people screened each record? Since Arksey & O’Malley (2005), convention has been that all records should be screened by at least two people. (See Robson, Pham, Hwee, Thomas, Rios, Page, & Tricco, 2019, for a summary of findings on the comparative accuracy of one vs. two reviewers. Single reviewers miss relevant studies.) |
We added the information in the text. Three people screened the records. |
|
4. Please be clear as to whether at least two researchers individually extracted data from each included publication. This is best practice according to JBI guidance for scoping reviews (Pollock, Peters, Khalil, McInterney, Alexander, Tricco, et al., 2023), however having two reviewers vs. one extract data may be less critical than for record screening (Robson et al., 2019). |
One researcher extracted the data and two other researchers doublechecked it. |
|
5. Was any other mechanism used for identifying records? For example, forward and ancestry citation searching, manual searching of journals, or suggestion of additional sources by the Africa Health Literacy Network with whom the authors consulted? There is no indication of this on the PRISMA diagram. Since the publication of PRISMA 2020 this has been considered best practice. |
Thank you for this recommendation. We intended to do so but due to the large number of articles found it was not possible to do so. Rather than narrowing down the scope of the article to go more in-depth with the search as explained by the reviewer, we focused on the references we had already identified. A comment is now added to the limitation paragraph in the discussion about this point. |
Reviewer 3 Report
Comments and Suggestions for Authors
This study provides a review of health literacy across Africa. The researchers provided a scoping review of 336 articles retrieved from six databases following the PRISMA guidelines. The sources include diverse materials like reports, policy briefs, book chapters, and academic publications. The analysis spans 38 of the 54 African countries, with most publications emerging in the last decade, dating back to 2001. It focuses on 13 key themes, including mental health, communicable and non-communicable diseases, maternal health, digital health, prevention, and health promotion, among others. The review also explores the actors and settings involved, revealing that health literacy efforts primarily occur in healthcare and community settings. This is an important review as there has little attention to health literacy in Africa as compared to Western countries.
However, I do have a few questions about the method which need to be clarified in the manuscript.
1. Please indicate whether a protocol exists for this review and if so, where. (See PRISMA guidelines.)
2. You have mentioned that the search was done across six online scientific databases. Did the authors consult a librarian to set up the search, or is one among the team members an expert in information science? If so, it should be mentioned.
3. How many people screened each record? Since Arksey & O’Malley (2005), convention has been that all records should be screened by at least two people. (See Robson, Pham, Hwee, Thomas, Rios, Page, & Tricco, 2019, for a summary of findings on the comparative accuracy of one vs. two reviewers. Single reviewers miss relevant studies.)
4. Please be clear as to whether at least two researchers individually extracted data from each included publication. This is best practice according to JBI guidance for scoping reviews (Pollock, Peters, Khalil, McInterney, Alexander, Tricco, et al., 2023), however having two reviewers vs. one extract data may be less critical than for record screening (Robson et al., 2019).
5. Was any other mechanism used for identifying records? For example, forward and ancestry citation searching, manual searching of journals, or suggestion of additional sources by the Africa Health Literacy Network with whom the authors consulted? There is no indication of this on the PRISMA diagram. Since the publication of PRISMA 2020 this has been considered best practice.
Analysis of findings is nicely done and neatly presented. I appreciate the author’s work, especially as it focuses on the African continent, as the study and development of health literacy can bring about change in the African continent. However, these methodological questions are important to determining the reliability of the findings. I can’t recommend publication without full information about them.
Author Response
Thank you for the helpful comments. We are pleased to provide the response below.
Best regards
Kristine Sørensen
Reviewer 3.
|
Lines 106 to 109 - explain why you only included articles in English and French; by excluding other languages widely spoken in Africa (e.g. Portuguese) you may be leaving out significant studies for the topic in question. In line 285 you mention that there is little contribution from Portuguese-speaking countries, but you didn't include that language in the study... (lines 339 - 340.) I suggest that this limitation be clearly presented and indicate what weaknesses it brings to the article. |
An initial test search showed no articles available in Portuguese, so we did not include in the search string. This point is added to the manuscript. Also we added that it was out of the scope to search on other languages such as Swahili, Afrikaans, etc. Eventually, all references found were in English since we used these data bases which presents references in English such as PubMed, PsycINFO, Eric, Cochrane and the African Journals Online. We appreciate this important advice and have added the language limitation as a weakness in the discussion of the method and tried to clarify the language decisions in the text. |
|
Lines 109 to 112 - the exclusion criteria indicate that countries or regions that are not internationally recognized have been excluded, couldn't this be introducing an error into the study? and why have you done this? aren't there issues that need to be reflected on these territories? I suggest explaining the need for ‘international recognition’ more clearly than what is presented in lines 325 - 332. |
For managerial reasons we made a choice to focus on all the countries in the African Union. We have now added that this can induce a bias in the dataset. We already acknowledged and described in the text the importance of ethnic and cultural factors across country boards and populations and that this should be taken into account in future studies. |
|
Lines 121-123 - I suggest explaining ‘They coded the data in Excel using a jointly developed coding scheme that reflected the research objectives’ what did this ‘jointly developed code’ consist of? |
We have specified the coding scheme which followed our research questions and how it was applied. |
|
Lines 138 - 143 - You should elaborate on how you analyzed the qualitative studies. As well as reflect on the constraints of relying primarily on quantitative studies (lines 273 - 274). |
We have specified the analytical process which included an account of qualitative and quantitative studies as well as other types of references included. |
|
Lines 155 - 163 - I suggest that a critical analysis should be made of the results obtained with regard to geographical distribution, can you reflect on the dominance found in studies carried out in South Africa and Nigeria, what factors could be responsible for this? academic factors? greater perception of the importance of health literacy? (more in-depth on lines 282-283) |
Apart from the reasons already mentioned in the text such as long-term collaboration with universities in the Global North and larger populations, colonial history and culture has been added as a possible influence. |
|
Lines 172 - 173 - isn't this a weakness? Why did you include these types of articles? |
We regard it as a strength as it was a deliberate choice to focus on scientific articles in general, not research publications only. The aim was to provide a general overview of the scope and scale of health literacy developments in research, policy and practice as presented in scientific literature. Hence, in our opinion we chose a method fitting our research question. We strongly believe that this kind of ‘helicopter view’ publication is missing for the region. The publication shows how much the health literacy developments have progressed and it should be known to the world through this ‘baseline’ publication. |
Round 2
Reviewer 3 Report
Comments and Suggestions for Authors
I am satisfied with the improvements the authors have made. I recommend publication.